# Socioeconomic determinants of *Schistosoma mansoni* infection using multiple correspondence analysis among rural western Kenyan communities: Evidence from a household-based study

Isaiah Omondi[1]☯*, Maurice R. Odiere[1]☯, Fredrick Rawago[1]‡, Pauline N. Mwinzi[1]‡, Carl Campbell[2]‡, Rosemary Musuva[1]☯

**1** Neglected Tropical Diseases Branch, Center for Global Health Research, Kenya Medical Research Institute (KEMRI), Kisumu, Kenya, **2** Center for Tropical and Emerging Global Diseases and Department of Microbiology, University of Georgia, Athens, Georgia, United States of America

☯ These authors contributed equally to this work.
‡ FR, PNM and CC also contributed equally to this work.
* iomondi19@gmail.com

## Abstract

### Background

Socioeconomic inequality including wealth distribution is a barrier to implementation of health policies. Wealth distribution can be measured effectively using household data on durable assets. Compared to other methods of analysing Socio-economic Status (SES) using durable assets, Multiple Correspondence Analysis (MCA) can create more reliable wealth quintiles. We therefore evaluated socioeconomic determinants of *Schistosoma mansoni* using MCA on household data among adult population in western Kenya. The hypothesis of this study was that MCA would be a useful predictor of *S. mansoni* prevalence and/or intensity.

### Methodology

Twelve villages, 6 villages that had showed the greatest decrease in *S. mansoni* prevalence (Responder villages) and 6 villages that showed relatively lower decrease (Hotspot villages) between the year 2011 and 2015 were randomly selected for this study. This was according to a previous Schistosomiasis Consortium for Operational Research and Elimination (SCORE) report from western Kenya. From each village, convenience sampling was used to identify 50 adults from 50 households for inclusion in this study. An interview with a questionnaire based upon MCA indicators was conducted. One stool sample from each of the 600 adults was examined based on four slides for *S. mansoni* eggs using Kato Katz technique. Mean Eggs per gram(EPG) was calculated by taking the average of the readings from the four slides. A log binomial regression model was used to identify the influence of the various age-groups(<30 years, 30-60 years and >60 years), household size, wealth

**Data Availability Statement:** All relevant data are within the manuscript and its Supporting information files.

**Funding:** Financial support to conduct this study was received from University of Georgia Research Foundation, Inc. which was funded by the Bill & Melinda Gates Foundation. The funders had no role in the study design, data collection and analysis, decision to publish or preparation of the manuscript.

**Competing interests:** The authors have declared that no competing interests exist.

class, occupation, education status, main water supply, sex and sub-county of residence on *S. mansoni* infection. EPG was then compared across variables that were significant based on multivariate log binomial model analysis using a mixed model.

## Principal findings

Overall prevalence of *S. mansoni* was 41.3%. Significantly higher prevalence of *S. mansoni* were associated with males, those aged below 30 years, those who use unsafe water sources (unprotected wells, lakes and rivers), residents of Rachuonyo North, Hotspot villages and those earning livelihood from fishing. Only sex and household size were significant predictors in the multivariate model. Males were associated with significantly higher prevalence compared to the females (aPR = 1.37; 95% CI = 1.14–1.66). In addition, households with at least four persons had higher prevalence compared to those with less than four (aPR = 1.29; 95% CI = 1.03–1.61). However, there was no difference in prevalence between the wealth classes(broadly divided into poor and less poor categories). Intensity of infection (Mean EPG)was also significantly higher among males, younger age group, Rachuonyo North residents and Hotspot Villages.

## Conclusion

Socioeconomic status based on an MCA model was not a contributing factor to *S. mansoni* prevalence and/or intensity possibly because the study populations were not sufficiently dissimilar. The use of convenience sampling to identify participants could also have contributed to the lack of significant findings.

## Introduction

Schistosomiasis affects aproximately 207 million people globally [1] and this accounts for close to 1.9 million disability-adjusted life years (DALYs) annually [2]. Of the overall global cases, 93% (192 million) are reported within sub-Saharan Africa region [3] where the disease is prevalent especially among resource-contrained communities with limited access to safe drinking water and adequate sanitation [4]. A higher percentage (70%) of those at risk of schistosomiasis infection is estimated to be school going children aged between 6 to 14 years [5].

Although some studies have indicated schistosomiasis to be a social disease [6], studies on the association of socioeconomic conditions with schistosomiasis have been equivocal, with some indicating association while others indicate no effect. A cross-sectional study in Walukuba Division in Eastern Uganda found that wealth index was a significant risk factor for infection with *S. mansoni* [7]. Another study in North Western Tanzania that used Principal Component Analysis (PCA) to create wealth quintiles demonstrated that control interventions against schistosomiasis and intestinal worms contributed towards improvement in socio-economic status of a community population [8]. Other studies have also shown that persistent schistosomiasis prevalence among school children could be attributed to socioeconomic inequality [9].

In Kenya, the role of socioeconomic factors in schistosomiasis transmission has been examined by several studies. For example, a cross sectional study involving school children from eight primary schools in Mbita district in Western Kenya found no effect of socioeconomic factors on *S. mansoni* infection risk [10]. Therefore, additional studies on the influence of

socioeconomic factors are warranted, especially in high transmission settings such as Western Kenya that need an adaptable schistosomiasis control strategy that considers local differences in disease ecology.

Standard Socioeconomic Status (SES) measures employ monetary information, including income or consumption expenditures. This however poses a caveat since the collection of accurate income data calls for extensive resources for household surveys; for instance, provisions need to be made for households and individuals generating income from multiple sources. Although consumption or expenditure measures are relatively more reliable and easier to collect compared to income, they require extensive data collection which could be time-consuming and therefore extra cost implications. Due to resource constraints when measuring household income or expenditure in low-and middle-income countries (LMICs), adoption of other SES evaluation methods that could streamline variable requirement has been necessitated hence allowing rapid data collection. Instead of income or expenditure, data are collected for variables that capture living standards, for example household ownership of durable assets (e.g. cupboard, motorvehicle, radio), infrastructure and housing features (e.g. main source of water, types of latrines). The use of asset-based indices of evaluating wealth in LMICs rather than per capita expenditures has gained popularity in the past few years especially in instances where data on expenditures are inaccessible or too costly to obtain. Traditional methods of creating SES indices have proved to suffer from recall bias, seasonality and data collection time. When per capita expenditure data are missing, the use of an asset index can clearly provide useful guidance to the order of magnitude of wealth-differentials [11].

There exists a vast volume of studies that have recommended asset-based methods of creating SES index which tend to be superior over traditional methods which rely on income and consumption expenditures. Such methodologies include the Filmer-Pritchett PCA, polychoric PCA and MCA. PCA is computationally easier, uses data that can easily be obtained from household surveys and uses all the asset variables in reducing the dimensionality of the data and therefore is a valid method for describing SES differentiation within a population [12]. Filmer-Pritchett procedure for conducting PCA requires that the variable be divided into a set of dummy indicators [13]. However, this could lead to deterioration of performance and probably produce a model with lower explanatory power. Therefore, polychoric correlations PCA method tend to outperform Filmer-Pritchett PCA if the proportion of explained variance is of importance [14]. A study comparing the performance of Filmer-Pritchett PCA, Polychoric PCA and MCA found that although the results were identical, MCA accounted for a higher proportion of the total variance of the household assets variables as compared to the other two methods [15]. The MCA model is also able to take into account variables measured on quantitative scale which is not possible with ordinary and polychoric PCA [16]. MCA is a data analysis technique used to detect and represent underlying structures within several categorical dependent variables in a data set. It accomplishes this by representing data as points in a low-dimensional Euclidean space. The MCA procedure could thus be regarded as the counterpart of PCA for categorical data [17]. MCA could also be considered as an extension of simple correspondence analysis with regard to its applicability to a large set of categorical variables. MCA allows analysis of a set of observations construed by a set of both nominal and quantitative variables.

However, studies employing MCA to investigate socioeconomic disparities about *S. mansoni* are still lacking. Health inequality data, including *S. mansoni* prevalence are often collected in community-based surveys but rarely analysed from a socioeconomic point of view. Yet, such data if properly analysed would be necessary for monitoring health inequalities and the impact of schistosomiasis control interventions at microeconomic level. The aim of this study was to establish the relationship between household SES and inequalities on *S. mansoni*

prevalence in a schistosomiasis endemic area of rural Western Kenya. The current study analyzed the relationship between SES and risk of infection with *S. mansoni* while controlling for measures of exposure to *S. mansoni* such as water and sanitation facilities.

This study opted to use prevalence ratios(PR) instead of Odds Ratios(OR) since PRs resulting from log-binomial model tend to be easier to interpret compared to ORs produced by logistic regression since log-binomial model uses a log link function whereas logistic regression model uses a logit link function [18]. When compared to Mantel-Haenszel, Cox, Robust Poisson and logistic regression; log-binomial regression would also be the best method for estimating prevalence ratios for intermediate prevalence [19, 20]. In order to identify the best model, we used Schwarz Bayesian Criteria(SBC) since some studies have proven its superiority over other Information Criteria (AIC and BIC), Heuristic methods and Model diagnostics [21]. This study hypothesized that the distribution of *S. mansoni* within the Western Kenya population is influenced by socioeconomic factors.

## Materials and methods

### Study area, economy and population

This community-based cross-sectional survey was conducted in the first quarter of 2018 in 12 villages in 3 sub-counties of Rarieda and Bondo (Siaya County) and Rachuonyo North subcounty (Homabay County) in Western Kenya. Mbita district which has been proven to be a schistosomiasis-hotspot zone is located in Homabay County [22]. Siaya county lies between latitude N 0˚26′ to N 0˚18˚ and longitude E 33˚58′ and W 34˚33′ and shares Lake Victoria border towards the South with Homa Bay County. Both counties are in the *S. mansoni*—endemic area of western Kenya and share many physical and socioeconomic environmental features. They include a tropical climate with average annual rainfall ranging from 1226–1572 mm and temperature ranging from of 21.7˚C—22.5˚C. The sub-counties border Lake Victoria where *S. mansoni* is highly endemic [22, 23]. The main socio-economic activities in the area are fishing, subsistence farming and petty trade. The main source of water for domestic use among the inhabitants of these areas is Lake Victoria which is also the primary source of *S. mansoni* infection.

### Study design

Villages for inclusion into this study were randomly selected from villages that had participated in the SCORE studies. An example of such studies was a chemotherapy-based initiative that was undertaken to control schistosomiasis in Western Kenya region that implemented school-based as well as village-wide annual mass drug administration (MDA) in varying treatment levels for five consecutive years (2011–2015) for 150 Kenyan villages situated on or near the Eastern shore of Lake Victoria [24]. In that study, only villages that had *S. mansoni* prevalence of ≥25% at baseline were included. At the end of the five-year period, villages that experienced ≤20% reduction in prevalence were termed Persistent Hotspots while those that experienced ≥40% reduction in prevalence were termed Responder Villages [25]. In the current study, six of the Responder Villages and another six of the Persistent Hotspot villages were randomly selected for inclusion. Convenience sampling for interview was then used to identify 50 adults aged ≥18 years from 50 households in each of the 12 villages. In instances where there were more than one adult in a household, the household head was selected for participation in the study. The selected individual must have been residents in the respective villages for a period of more than a year and were knowledgeable about the culture and norms of the people. Informed consent was obtained from study participants prior to enrolment into the study. Pretested questionnaire was administered to the participants and a single stool and

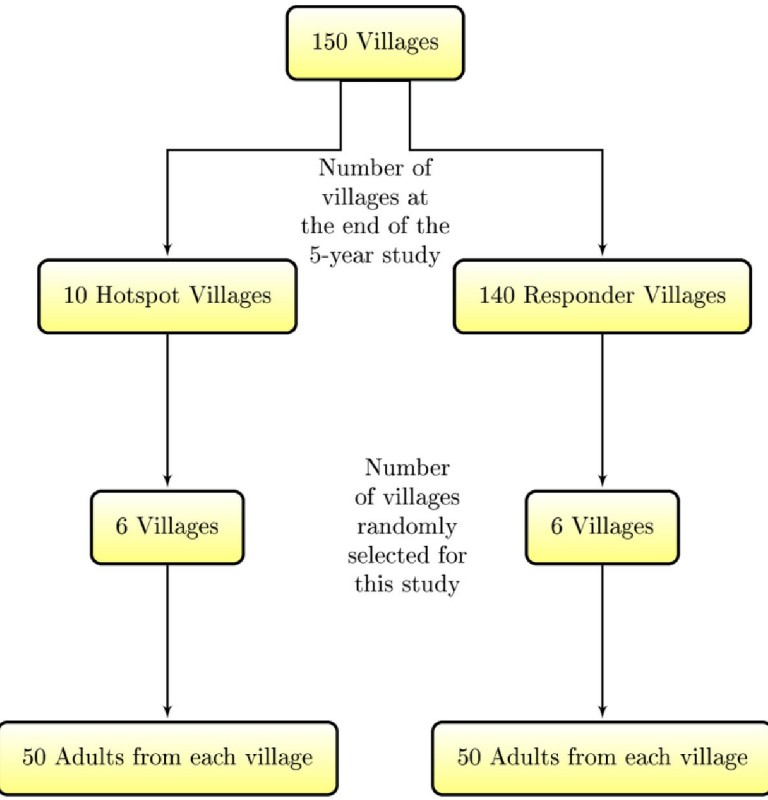

**Fig 1. Schematic diagram for sample selection for inclusion in the study.**

urine sample was collected for examination at the KEMRI laboratory. The process of sample selection has been illustrated in Fig 1.

## Parasitological assesment

Stool samples were collected from the 600 participants. Each participant was provided with a plastic container for a stool sample and plastic cup for urine prior to the day of survey and instructed to bring the samples during the survey. Parasitological assessment was based on one stool sample (4 slides) per participant. Each slide was analyzed by the Kato-Katz technique for eggs of *S. mansoni*, and soil-transmitted helminths (*Ascaris lumbricoides*, *Trichuris trichiura* and hookworm). 41.7 mg of faeces was used for each slide. Eggs were counted by two independent microscopists and any discrepancy in results of the two was reconciled by comparing to results of a third independent and more experienced microscopist. The average of the readings from two microscopists with the closest identical results formed the basis for calculating EPG. World Health Organization (WHO)—proposed thresholds were used to classify the intensity of infection [26]. Individuals who tested positive for *S. mansoni* were treated with praziquantel (PZQ), single dose of 40mg/kg, whereas those who tested positive for soil-transmitted helminths were treated with a single dose of albendazole (400mg). All statistical analysis in this paper were based on the *S. mansoni* prevalence and intensity from Kato-Katz technique. The urine sample was used to investigate *S. mansoni* using M-Reader technology and POC-*Schistosoma* circulating cathodic antigen(CCA) technology which were evaluated under the broader epidemiological project from which this study was extracted. The soil transmitted helminths

(STH) and urine results are not reported in this paper. The prevalence of Schistosoma haematobium was not assessed in this study.

## Collection of socioeconomic data

Socioeconomic data was collected from each participant who provided stool and urine samples for parasitological examination. Data on individual and household characteristics with a possible influence on schistosomiasis (sex, place of residence, age, source of water for domestic use, highest level of education attained, occupation of the household head, number of persons in the household, type of latrine used by the household members) was collected. In addition, information about factors indicating the family's wealth index (living in a brick or wood/mud house, existence of electricity and other assets such as refrigerator, sofa sets, dining table, car, tractor, cell phone, cupboard, wardrobe, motorcycle, bicycle) was collected through interviewing the household head using a structured interview with a questionnaire in English. In instances where the the respondent was not conversant with English, the same questionnaire was availed in the local language (majorly Dholuo). The questions were formulated according to standard questionnaires adjusted to local requirements and appropriateness. Data was captured real-time using a Commcare application installed in smart phones and submitted to a central server on completion of each entry. Plausibility was thereafter checked, and data cleaned before the database was locked. All information on participants and their respective households was treated confidentially.

## Multiple correspondence analysis

To obtain wealth indices and their scores, the following socioeconomic indicator variables were considered: Materials used for walls of main house (no bricks, unburnt bricks, burnt bricks with mud, burnt bricks/stones with cement); material used for the floor of main house (not cemented, cemented, tiled); materials used for roofing of main house (thatch, iron sheets, tiles); source of water (no tap water, tap water outside house, tap water inside house); sanitation facility (none, pit latrine, ventilated improved pit latrine, flush toilet); household assets (radio, cupboard, bicycle, paddle dhow, telephone, television, motorcycle, car, engine dhow, sewing machine, refrigerator, wall clock, sofa) and existence of electricity. Open sources of water such as rivers and lakes were considered unsafe while protected sources such as tap water were considered safe. All the options above were transformed into binary/indicator variables prior to running the MCA model, with 1 denoting presence and 0 denoting absence of the respective facility/asset. The scores generated by the MCA model formed the basis for classifying individuals into five socio-economic quintiles with the first quintile as the poorest and the fifth quintile as the least poor. Finally, a binary variable was generated with the first three SES quintiles (i.e., poorest, second and third poorest) combined to form the 'poorest' category and the fourth and fifth quintiles combined to form the 'less-poor' category. The 'less-poor' category was then adopted as the reference category in the log-binomial and mixed regression models.

## Statistical analysis

The *S. mansoni* data comprised records from 600 individuals from the study area. This study aimed at exploring not only the prevalence of *S. mansoni* but also its intensity. Shapiro-wilk test was used to test for normality of the mean EPG. The results indicated a violation of the normality assumption, hence, a log-transform of the mean EPGs was considered in order to attain normality. Several explanatory variables specific to the individuals, such as age, sex,

education level, occupation, wealth index, subcounty of residence, size of the household, main source of water and presence of latrine were considered in developing the final model.

This study used univariate and multivariate regression models to examine the association between the prevalence of *S. mansoni* and the wealth index. Given that *S. mansoni* is not a rare disease in Lake Victoria region, prevalence ratios were used to estimate risk of *S. mansoni* infection. First, the univariate prevalence ratios for the demographic, socioeconomic and sanitation variables were generated by considering regression models with a single predictor variable. A generalized score statistic formed the basis for determining the overall significance of each variable, whereas the p-values and confidence intervals formed the basis for evaluating statistically significant differences between variable levels. The level considered to be least associated with the risk for contacting *S. mansoni* was chosen as the reference category.

To develop the multivariable model, several plausible interactions such as wealth index and presence of latrine, wealth index and highest education level, wealth index and household size or wealth index and the main source of water for domestic use were assesed, adding each interaction to the full model containing all the predictors presented in the univariate analysis. The results indicated that none of these interactions were significant and moreover the parameter estimates observed in the main effects registered probable problems with multicollinearity possibly resulting from model burdening correlations between the variables. The estimates exhibited by a section of the variables resulted in prevalence ratios whimsical to the general understanding of the risk factor in question.

Among all the possible models, SBC was used to identify the best model. Several plausible interactions such as SES and presence of latrine, SES and highest education level, SES and household size or SES and the main source of water were included in the model containing all the independent variables. Among all the possible models, the model with the lowest value with respect to SBC was selected but none of the interactions was present in the final model. Then, confounding was assessed by adding each of the variables considered in the univariate analysis but not included in the final multivariable analysis to the selected model. The variables sex, sub-county and household size remained significantly associated with infection when controlling for the other variables in the model. Those variables not entered into the multivariable model as significant predictors for *S. mansoni* were assessed as confounders.

We then examined the intensity of infection by each of the independent variables included in the ultimate multivariate model. This analysis used the log-transformed geometric mean EPG among those individuals who tested positive for *S. mansoni*. The geometric mean and confidence intervals for each level of the predictors included in the multivariable model were then calculated using a mixed model. To achieve this, we modelled the $log_{10}$ of the mean EPG versus the covariates of interest, capturing the least squares mean and confidence interval, and then transforming the estimates back to the normal scale (rather than the log scale) of the EPGs when presenting the statistical analysis results. To investigate presence of any differences in the mean log EPG among the covariate levels, we used an overall F-test of the mixed model. Finally, we assessed significance in the pair-wise differences in means across all possible combinations of the covariate levels using a tukey multiple comparison adjustment [27]. Cases with missing stool lab results were excluded from analysis. All analyses were completed in SAS (version 9.4).

## Ethical considerations

The review and approval of this study was done by the Scientific and Ethical Review Committee of the Kenya Medical Research Institute (KEMRI). The household heads had to provide a

written informed consent before participating in the study. All participant data were de-identified and stored in a secure (password-protected) hard drive with restricted access.

# Results

## Parasitological results

Of the 600 study participants, 25.7% were males while 74.3% were females. The population age ranged from 18 to 90 years (mean 44.7, SD 16.5, and median 42.5 years), with 21.5%, 56.5% and 22% aged below 30 years, between 30 to 60 years and above 60 years, respectively. The overall prevalence of *S. mansoni* was 41.3% while the overall geometric mean egg count was 56.2 eggs per gram of feces. Of the *S. mansoni* positive cases, 61.9%, 28.1% and 10.0% had light, moderate and heavy infection intensities respectively. The prevalence of *S. mansoni* among those aged between 18 to 30 years, between 30 to 60 years and above 60 years was 48.1%, 41.9% and 33.3% respectively while the prevalence among males and females was 53.3% and 37.2%, respectively. In addition, the prevalence of *S. mansoni* within Hotspot and Responder villages was 48.67% and 34.33% respectively. The arithmetic mean EPG peaked at 114.4 eggs per grams of feces among males and at 105.6 eggs per grams of feces for those aged below 30 years old.

## Associations between infection and variables

The most notable results were in the age group below 30 years where prevalence of *S. mansoni* was 1.44(95% CI = 1.07–1.95) times higher than the prevalence in those above 60 years old as a reference group; the prevalence among males was 1.43(95% CI = 1.18–1.73) times higher compared to the prevalence among females; whereas the prevalence among residents of Rachuonyo North sub-county was 1.51(95% CI = 1.11–2.06) times higher than the prevalence of those from Bondo sub-county. In addition, the prevalence in Hotspot villages was 1.42(95% CI = 1.17–1.72) times higher when compared to the prevalence in Responder villages (Table 1).

Among the socio-economic and sanitation variables considered, occupation, the source of water for domestic use and the size of the households were significantly associated with prevalence of *S. mansoni*. However, there was no significant association between the level of

**Table 1. Prevalence Ratios (PR) and 95% Confidence Intervals (CI) for *S. mansoni* infection of individuals to demographic variables.**

| Variable | No. of individuals | Infected n (%) | PR | 95% CI | p-value |
|---|---|---|---|---|---|
| Sex | | | | | |
| Male | 154 | 82(53.25) | 1.43 | 1.18–1.73 | 0.0002 |
| Female | 446 | 166(37.22) | 1.00 | | |
| Age group | | | | | |
| <30 | 129 | 62(48.06) | 1.44 | 1.07–1.95 | 0.0170 |
| 30–60 | 339 | 142(41.89) | 1.26 | 0.96–1.65 | 0.0996 |
| >6 | 132 | 44(33.33) | 1.00 | | |
| Sub-county of residence | | | | | |
| Rachuonyo North | 300 | 145(48.33) | 1.51 | 1.11–2.06 | 0.0088 |
| Rarieda | 200 | 71(35.50) | 1.11 | 0.79–1.56 | 0.5512 |
| Bondo | 100 | 32(32.00) | 1.00 | | |
| Type of Village of residence | | | | | |
| Hotspots | 300 | 146(48.67) | 1.42 | 1.17–1.72 | <0.001 |
| Responders | 300 | 103(34.33) | 1.00 | | |

**Table 2. Prevalence Ratios (PR) and 95% Confidence Intervals (CI) for *S. mansoni* infection of individuals according to socioeconomic and sanitation characteristics.**

| Variable | No. of individuals | Infected n (%) | PR | 95% CI | p-value |
|---|---|---|---|---|---|
| Education level | | | | | |
| None | 282 | 113(40.07) | 0.94 | 0.78–1.14 | 0.5549 |
| Atleast Primary | 318 | 135(42.45) | 1.00 | | |
| Occupation | | | | | |
| Fishing | 94 | 57(60.64) | 1.54 | 1.24–1.92 | 0.0001 |
| Agriculture | 229 | 82(35.81) | 0.91 | 0.73–1.14 | 0.4150 |
| All other occupations | 277 | 109(39.35) | 1.00 | | |
| Socio-economic status | | | | | |
| Poor | 360 | 142(39.44) | 0.89 | 0.74–1.08 | 0.2468 |
| Less Poor | 240 | 106(44.17) | 1.00 | | |
| Water supply | | | | | |
| Unsafe | 375 | 169(45.07) | 1.28 | 1.04–1.58 | 0.0197 |
| Safe | 225 | 79(35.11) | 1.00 | | |
| Household size | | | | | |
| At least 4 persons | 416 | 187(44.95) | 1.36 | 1.08–1.71 | 0.0098 |
| <4 persons | 184 | 61(33.15) | 1.00 | | |
| Latrine present? | | | | | |
| No | 136 | 55(40.44) | 0.97 | 0.77–1.22 | 0.8111 |
| Yes | 464 | 193(41.59) | 1.00 | | |

education, wealth class, and presence of latrine in the respective household and the prevalence of *S. mansoni*. The prevalence among those individuals whose main source of livelihood was fishing was 1.54 (95% CI = 1.24–1.92) times higher than the prevalence among those whose main source of livelihood was other occupations (apart from fishing and farming). However, there was no difference in prevalence between those who have farming and those who have other occupations as the main source of earning a living. *S. mansoni* prevalence among individuals who used unsafe water sources as the main source of water for domestic use was 1.28 (95% CI = 1.04–1.58) times higher compared to the prevalence among those who used safe water sources as the main source of water for domestic use. With regard to housing conditions, the prevalence for individuals who came from households with at least four persons was 1.36 (95% CI = 1.08–1.71) times the prevalence for those who came from households with less than four persons. However, education level, SES and presence of latrine in the household where the individual came from were not significantly associated with *S. mansoni* infection (Table 2).

Table 3 shows the results of the multivariable log binomial model for *S. mansoni* infection. Age was only significant in the model where education level and occupation were not included, thus we concluded that the association between age and *S. mansoni* infection was confounded by occupation and education level. However, SES remained non-significant even after including education level, occupation or presence/absence of latrine in the final model. Holding constant the other predictors entered into the model, sex was strongly associated with *S. mansoni* infection. Males were 1.37(95% CI = 1.14–1.66) times more likely to be infected when compared with females. The other variables or variable levels showing a significant or marginally significant association with infection when compared with the referent level were sub-county of residence, age-group, number of persons per household and type of village (Hotspot/Responder). Residents of Rachuonyo North sub-county were 1.48(95% CI = 1.09–2.01) times more likely to be infected when compared to those from Bondo sub-county;

**Table 3. Multivariate log binomial model of the association between demographic, socioeconomic and sanitation variables and *S. mansoni* prevalence.**

| Variable | No. of individuals | Adj. PR | 95% CI | p-value |
|---|---|---|---|---|
| Sex | | | | |
| Males | 154 | 1.37 | 1.14–1.66 | 0.0010 |
| Females | 446 | 1.00 | | |
| Sub-county of residence | | | | |
| Rachuonyo North | 300 | 1.48 | 1.09–2.01 | 0.0131 |
| Rarieda | 200 | 1.23 | 0.87–1.73 | 0.2430 |
| Bondo | 100 | 1.00 | | |
| Socio-economic status | | | | |
| Poor | 360 | 0.92 | 0.77–1.11 | 0.3770 |
| Less Poor | 240 | 1.00 | | |
| Water supply | | | | |
| Unsafe | 375 | 1.19 | 0.96–1.47 | 0.1171 |
| Safe | 225 | 1.00 | | |
| Household size | | | | |
| At least 4 persons | 416 | 1.29 | 1.03–1.61 | 0.0300 |
| <4 persons | 184 | 1.00 | | |
| Age group | | | | |
| <30 | 129 | 1.36 | 1.01–1.82 | 0.0419 |
| 30–60 | 339 | 1.22 | 0.93–1.60 | 0.1549 |
| >60 | 132 | 1.00 | | |
| Type of Village of residence | | | | |
| Hotspots | 300 | 1.17 | 1.13–1.19 | <0.001 |
| Responders | 300 | 1.00 | | |

individuals aged below 30 years were 1.36(95% CI = 1.01–1.82) times more likely to be infected as compared to those in the above 60 years; households with at least four persons were 1.29 (95% CI = 1.03–1.61) times more likely to be infected as compared to those with more than four persons while residents of Hotspot villages were 1.17(95% CI = 1.13–1.19) times more likely to be infected as compared to those from Responder villages. Wealth index was neither significant in the univariate analysis (Table 2) nor in the multivariate analysis (Table 3).

Geometric mean egg counts were significantly associated with sex (F-statistic = 9.08, p = 0.0029), sub-county of residence (F-statistic = 14.19, p<0.001), age (F-statistic = 4.19, p = 0.0163) and type of village of residence (F-statistic = 9.21, p = 0.0028). The tukey multiple comparison procedure indicated that geometric mean egg counts among Rachuonyo North residents was significantly higher as compared to that among Rarieda sub-county residents. In addition, the mean EPG among Rarieda residents was also significantly low when compared to that among Bondo sub-county residents although the EPGs were not significantly different when Rachuonyo North was compared with Bondo. In comparing geometric EPGs among below 30 years, 30–60 years and above 60 age-groups, the tukey multiple procedure indicated that the EPG among those aged below 30 was significantly higher as compared to those aged between 30–60 years and above 60 years. However, there was no significant difference in EPG among those aged between 30–60 when compared to those aged >60 years. The EPG among the residents of Hotspot villages was significantly higher compared to residents of Responder villages. The overall statistics did not indicate any significant differences in the geometric mean egg counts with type of water supply (safe/unsafe), SES or household size (Table 4).

**Table 4. Intensity of *S. mansoni* infection of individuals expressed as geometric mean of *S. mansoni* eggs per gram of feces, by predictors in the multivariable model.**

| Variable | No. Infected | Mean EPG ($log_{10}$) | Geometric Mean EPG | 95% CI | Turkey grouping* |
|---|---|---|---|---|---|
| Sex | | | | | |
| Males | 82 | 4.24 | 80.55 | 57.31–113.23 | A |
| Females | 166 | 3.65 | 47.28 | 37.81–59.11 | B |
| Overall F-statistic = 9.08; p = 0.0029 | | | | | |
| Sub-county of residence | | | | | |
| Rachuonyo North | 145 | 4.46 | 82.27 | 63.83–106.03 | A |
| Rarieda | 71 | 3.65 | 24.64 | 18.32–33.15 | B |
| Bondo | 32 | 4.02 | 59.47 | 40.58–87.16 | A |
| Overall F-statistic = 14.19; p = <0.001 | | | | | |
| Socio-economic status | | | | | |
| Poor | 142 | 3.89 | 53.47 | 41.39–69.06 | A |
| Less Poor | 106 | 4.02 | 60.18 | 45.45–79.69 | A |
| Overall F-statistic = 0.44; p = 0.5066 | | | | | |
| Water supply | | | | | |
| Unsafe | 169 | 3.99 | 58.56 | 46.65–73.53 | A |
| Safe | 79 | 3.90 | 51.51 | 36.59–72.52 | A |
| Overall F-statistic = 0.20; p = 0.6546 | | | | | |
| Household size | | | | | |
| At least 4 persons | 187 | 3.96 | 55.36 | 44.66–68.64 | A |
| <4 persons | 61 | 3.94 | 58.98 | 39.50–88.06 | A |
| Overall F-statistic = 0.01; p = 0.9132 | | | | | |
| Age group | | | | | |
| <30 | 62 | 4.37 | 77.82 | 53.29–113.63 | B |
| 30–60 | 142 | 3.92 | 56.38 | 44.04–72.18 | A |
| >60 | 44 | 3.55 | 34.36 | 21.91–53.89 | A |
| Overall F-statistic = 4.19; p = 0.0163 | | | | | |
| Type of Village of residence | | | | | |
| Hotspots | 146 | 4.28 | 72.44 | 57.04–92.00 | A |
| Responders | 103 | 3.67 | 39.26 | 29.23–52.73 | B |

* The mean EPG for various categories of each variable were tested for significant differences. Categories indicating no statistically significant difference within a variable display the same letter while those that are significantly different do not have the same letter.

## Discussion

This cross-sectional study conducted in rural settings proximate to Lake Victoria which had been associated with high endemicity of schistosomiasis in Western Kenya according to previous studies [22, 28, 29] evaluated the relationship between burden of *S. mansoni* infection and household SES index. Our findings indicate no difference in prevalence of *S.mansoni* between individuals in the poorest households compared to those from less-poor households. Among the socioeconomic indicators considered (wealth class, occupation, education level and household size), only household size and occupation turned out to be significant predictors of *S. mansoni* occurrence in both the univariate and multivariate models.

Prevalence of *S. mansoni* was significantly higher in Rachuonyo North sub-county compared to Rarieda and Bondo sub-counties. This could possibly attributed to higher number of intervention programs executed in Bondo and Rarieda in the recent past hence leading to a remarkable decline in prevalence. These differences could also possibly be attributed to the

focal distribution of *S. mansoni* and closer proximity of villages in Rachuonyo North compared to Rarieda and Bondo sub-counties to Lake Victoria.

The association between number of persons per household and *S.mansoni* infection corroborates with the findings from a study conducted in Brazil which found out that as the number of persons per room increased, the probability of schistosomiasis occurrence also increased, although this association was only marginal [30]. Findings from our analysis, however, contrast with those of other studies conducted in rural Yemen and Nigeria both of which found no association between household size and *S. mansoni* infection [31, 32]. The Yemen-based study, however, was limited to children aged ≤15 years while the Nigeria-based study involved both children and adults (aged 1–90 years) whereas in the Brazil-based study the analysis was not restricted to any specific age group (based on both children and adults (aged 2–95 years)). In addition, both studies in Yemen and Nigeria were based on populations in which *S.mansoni* was relatively low, that is, 9.3% and 17.8%, respectively. However both studies used odds ratios from logistic regression to estimate the prevalence ratios unlike both the current study and Brazil-based study which used prevalence ratios from a log binomial regression model.

With regard to the association between *S. mansoni* prevalence and wealth classes we found no significant association in prevalence between socioeconomic groups. These findings are consistent with the results from a study conducted in Tanzania that used PCA to calculate the quintiles [9]. Our results however contrast the results from a study conducted in Jinja District, Uganda and a systematic review that comprised of publications from 20 countries [7, 33] both of which found out that wealth index is an independent predictor of schistosomiasis occurrence. The former study generated wealth classes by assigning arbitrary scores to household assets, for example, materials used for the walls of main house (no bricks = 1, unburnt bricks = 2, burnt bricks with mud = 3, burnt bricks, stones with cement = 4). This could however at times generate subjective classes when applied to other settings. Other studies that found a significant association between SES and schistosomiasis prevalence include those conducted in Yemen and Nigeria (mentioned above) which relied on individual income levels to create SES indices. This mode of generating SES indices could at times suffer from recall bias, seasonality and variation depending on the timing for data collection activity [31, 32]. Our findings may be attributed to the poor socioeconomic environment in villages proximate to Lake Victoria. The absence of socioeconomic effect could be attributed to the widespread poverty and the rural way of life which tend to put the whole community at risk of infection.

The current study found a significant association between occupation and *S. mansoni* whereby those whose main source of livelihood was fishing were at greater risk of being infected with *S. mansoni* compared to those whose main source of livelihood was farming among other occupational activities. This could be due to possible contact with infested water in the course of duty. Some previous studies have also pointed out occupation to be a significant predictor of schistosomiasis [30, 34, 35]. The first study was only based on pregnant women aged ≥15; the second study did not involve any specific age bracket (population aged between 0–96) while third study considered the possibility of multiple sampling of individuals from the same household by analyzing correlation structures between samples obtained from such individuals.

No significant differences in *S.mansoni* prevalence between individuals who possessed atleast primary education and those who were less educated was observed in this rural Western Kenya community. These findings are in agreement with those from a study conducted in North Central Nigeria that involved both children and adults [32], and a cross-sectional community-based field study conducted among pregnant women in Nigeria [34]. The findings of the current study however contrast the findings of a cross-sectional community-based study carried out among children aged ≤15 years in rural areas in Yemen [31]. Our findings also

contrast the findings of a systematic literature review on the socioeconomic distribution within endemic countries of *lymphatic filariasis*(LF), onchocerciasis, schistosomiasis, STH, trachoma, Chagas' disease, Human African Tripanosomiasis(HAT), leprosy, and *visceral leishmaniasis* (VL) which involved publications between 2004–2013 in countries comprising almost 90% of the global burden for the studied Neglected Tropical Diseases(NTDs) [33]. In addition our findings contrast those of a study conducted in a low range of medium prevalence for schistosomiasis in South East Brazil which also involved both children and adults [35].

Among the demographic variables considered, our study identified sex to be the strongest predictor of *S.mansoni* infection. This association has also been identified by other studies in Nigeria and Brazil. [32, 35].

## Limitations

We acknowledge some limitations of our methodology. This study had to rely on a single faecal sample collection instead of the ideal three consecutive samples and multiple Kato-Katz smears examination [36]. Therefore, the prevalence and intensity of *S. mansoni* were likely to be underestimated/overestimated due to the temporal variation in egg excretion over hours and days. Convenience sampling was used to identify participants for interview yet this method could result in under-representation or over-representation of particular groups within the sample. Moreover, our findings were based on a single cross-sectional survey hence the inability to evaluate cause-and-effect of SES on *S. mansoni* prevalence over time. A longitudinal or trend analysis of repeated surveys would be ideal in analysing changes in SES and monitoring the gap in *S. mansoni* prevalence between the poorest and less-poor households.

## Conclusion

This study did not find a significant association between *S. mansoni* infection and socioeconomic status. This could possibly be attributed to the widespread poverty in the study area. *S. mansoni* prevalence and intensity were significantly higher in Hotspot villages, among males, in younger age group and in Rachuonyo North subcounty.

## Supporting information

**S1 Data. Manuscript data.** Data on which the analysis was based.
(XLS)

**S2 Data. Data dictionary.**
(XLSX)

## Acknowledgments

We express our gratitude to all interviewees for their participation in this study. We are also grateful for the continuous endeavours of fieldworkers and laboratory technicians of the Kenya Medical Research Institute (KEMRI) NTD branch without whose efforts this research would not have been a success, and to the members of the Health Department of Siaya and Homabay Counties for their enduring collaboration.

## Author Contributions

**Conceptualization:** Isaiah Omondi, Rosemary Musuva.

**Data curation:** Isaiah Omondi.

**Formal analysis:** Isaiah Omondi.

**Funding acquisition:** Rosemary Musuva.

**Investigation:** Isaiah Omondi, Maurice R. Odiere, Rosemary Musuva.

**Methodology:** Rosemary Musuva.

**Resources:** Rosemary Musuva.

**Supervision:** Pauline N. Mwinzi.

**Writing – original draft:** Isaiah Omondi.

**Writing – review & editing:** Isaiah Omondi, Maurice R. Odiere, Fredrick Rawago, Pauline N. Mwinzi, Carl Campbell, Rosemary Musuva.

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
