## [Decision Letter · Decision Letter 0]

31 Jul 2020

PONE-D-20-05468

Socioeconomic determinants of schistosomiasis infection using multiple correspondence analysis among rural western Kenyan communities: evidence from a schistosomiasis household-based research.

PLOS ONE

Dear Dr. ISAIAH OMONDI,

Thank you for submitting your manuscript to PLOS ONE. After careful consideration, we feel that it has merit but does not fully meet PLOS ONE’s publication criteria as it currently stands. Therefore, we invite you to submit a revised version of the manuscript that addresses the points raised during the review process.

We look forward to receiving your revised manuscript.

Kind regards,

Mary Hamer Hodges

Academic Editor

PLOS ONE

Journal Requirements:

2.We noticed you have some minor occurrence of overlapping text with the following previous publications, which needs to be addressed:

* Vyas, Seema, and Lilani Kumaranayake. "Constructing socio-economic status indices: how to use principal components analysis." Health policy and planning 21.6 (2006): 459-468.

* Were, Vincent, et al. "Socioeconomic health inequality in malaria indicators in rural western Kenya: evidence from a household malaria survey on burden and care-seeking behaviour." Malaria journal 17.1 (2018): 166.

* Gazzinelli, Andrea, et al. "Socioeconomic determinants of schistosomiasis in a poor rural area in Brazil." Acta tropica 99.2-3 (2006): 260-271.

 In your revision ensure you cite all your sources (including your own works), and quote or rephrase any duplicated text outside the methods section. Further consideration is dependent on these concerns being addressed.

3.We note that you have indicated that data from this study are available upon request. PLOS only allows data to be available upon request if there are legal or ethical restrictions on sharing data publicly. For information on unacceptable data access restrictions, please see http://journals.plos.org/plosone/s/data-availability#loc-unacceptable-data-access-restrictions.

Reviewers' comments:

Reviewer's Responses to Questions

**Comments to the Author**

1. Is the manuscript technically sound, and do the data support the conclusions?

Reviewer #1: Partly

Reviewer #2: Yes

2. Has the statistical analysis been performed appropriately and rigorously? 

Reviewer #1: Yes

Reviewer #2: Yes

3. Have the authors made all data underlying the findings in their manuscript fully available?

Reviewer #1: No

Reviewer #2: No

4. Is the manuscript presented in an intelligible fashion and written in standard English?

Reviewer #1: No

Reviewer #2: Yes

5. Review Comments to the Author

Reviewer #1: Abstract

1. There are a number of minor typos which can easily be corrected. Nevertheless, there are too many long sentences through the manuscript with mixed up information which need to be fixed.

2. Need paragraphing

Introduction

1. Need paragraphing

2. Too many long sentences

3. Grammatical errors

4. Schistosomiasis includes both mansoni and haematobium

Methods

1. Some terms need either defining or referencing. (check the annotated PDF file)

2. Urine samples collected. However, what happens has not been said. This needs to be clarified.

3. Needs major revisions

4. What was the basis for having three age groups? >30, 30-60....what is the upper age limit for the first group and lower limit for the second group? This needs clarifying.

Results

1. A lots of sections needs to be improved. The methods sections have been mixed with results. In this section reports results only.

Discussion

1. Some section needs supporting literature.

2. Too long sentences

Conclusion

1. Needs to be re-aligned with the objective of the study

Limitations

1. Move limitations to before conclusion.

Reviewer #2: 1. What is the reason for excluding results from the urine samples collected from the analysis?

2. Please specify which water sources were considered safe and unsafe in the methods section

3. Would you expect the variables to change significantly if results from the urine analysis would have been considered?

6. PLOS authors have the option to publish the peer review history of their article (what does this mean?). If published, this will include your full peer review and any attached files.

Reviewer #1: No

Reviewer #2: No

---

## [Author Response · Author response to Decision Letter 0]

23 Sep 2020

Dear Sir/Madam

I'm very grateful for your continued support since I submitted my manuscript to PLOS ONE for review and eventual publishing. Below are the responses to the issues you raised in the last review:

1. Information on financial disclosure has been updated

2. Authors and their affiliations has been updated to the required format

3. The occurrence of overlapping text with previous publications have been eliminated

4. Data has been made available after having been de-identified

5. Instances of ambiguous texts have been minimized

6. Paragraphing has been introduced in the Abstract section

7. The long sentences have been split and grammatical errors editted

8. Schistosomiasis has been specified to mean schistosoma mansoni

9. Clarification has been made with regard to urine samples that they were used for another component of the main study

10. The age-groups have been specified. The lower age-limit was 18 years

11. Some section of the results section have been moved to the methods section

12. Rephrasing and rewording has been done on the Conclusion section to align it with the objective of the study

13. Limitations section has been moved before Conclusion

14. A distinction has been made as to what constitutes safe and unsafe sources of water

Thank you

Isaiah Omondi

---

## [Decision Letter · Decision Letter 1]

19 Oct 2020

PONE-D-20-05468R1

Socioeconomic determinants of schistosomiasis infection using multiple correspondence analysis among rural western Kenyan communities: evidence from a schistosoma mansoni household-based research.

PLOS ONE

Dear Dr. ISAIAH OMONDI

Thank you for submitting your manuscript to PLOS ONE. After careful consideration, we feel that it has merit but does not fully meet PLOS ONE’s publication criteria as it currently stands. Therefore, we invite you to submit a revised version of the manuscript that addresses the points raised during the review process.

The current use of S. mansoni must be addressed throughout the manuscriptThe text is not clear regarding the sample size 50 per stratum is illustrated by then 600 stool samples presentedNo reference to S. heamatobium results despite describing urine collectionNo comparision of Hotspots vs RespondersSpecific feedback from my evaluation of the manuscript has been provided

We look forward to receiving your revised manuscript.

Kind regards,

Mary Hamer Hodges, MBBS MRCP DSc

Academic Editor

PLOS ONE

Additional Editor Comments (if provided):

Your revision is still in need of both minor and major improvements. I have itemized these in detail and made a number of suggestions for your consideration:

Current title:

Socioeconomic determinants of schistosomiasis infection using multiple correspondence analysis among rural western Kenyan communities: evidence from a schistosoma mansoni household-based research.

Suggested title:

Socioeconomic determinants of Schistosoma mansoni infection using multiple correspondence analysis among rural western Kenyan communities: evidence from a household-based study.

Throughout this article the term schistosomiasis is still being used instead of S. mansoni. The full term Schistosoma mansoni should be used in the title and the first time in the Abstract and in the main text after that it should be abbreviated to S. mansoni.

Lines that need to be corrected include

Abstract Line 6, All table titles 1-4.

L76, 84,85,86, 97,102, 106, 116, 287, 305, 418, 438, 476, 478, 449 (S.Mansoni)

Please note the ‘S’ should be capitalized, the ‘M’ should not be capitalized, on several occasions the lower case ‘s’ or the upper case ‘M’ was used.

ABSTRACT

BACKGROUND

Line 5 (MCA)can without a space should read (MCA) can

Please state what was the original hypothesis? Presumably that MCA would be a useful predictor of S. mansoni prevalence (and/or intensity)?

METHODS

Line 3 ‘over the years’. How many years?

You describe a sample size of 600 but your Figure 1 in the main text describes 50 adults per stratum. Please explain. Are the 50 adults only those interviewed? If so please edit the title of Figure 1.

The main text describes taking urine as well as stools but no mention is made of those findings for S. haematobioum.

FINDINGS

Where is the comparison by strata: Hotspots vs Responders?

I don’t think you do justice to your findings by only reporting on the significances by prevalence using the multivariate analysis. I think the general reader will want to put that into perspective by understanding the significant higher prevalence in males, younger age group, unsafe water source, region and occupation (fishing).

You make no mention of significant findings by intensity of S. mansoni infection which was higher in males, younger age group and region.

When stating the lack of significant by prevalence it is really important to draw the readers attention to the wealth quintiles only spanning poor and less poor.

CONCLUSION: This need to relate to the original hypothesis such as:

The MCA model was not an effective predictor of S. mansoni prevalence and or intensity. Perhaps the study populations were not sufficiently dissimilar?

These comments refer all to the main text which would be better presented in a more concise manner. In the METHODS Line 173-181 seem better suited to BACKGROUND.

Minor comments:

Line 120 ‘other’ should be replaced by ‘another’

Line 121 ‘Convenience’ sampling for what stool sampling or interview?

Line 125 in missing a ‘.’

Line 147 Urine should be urine

FINDINGS

No S. haematobium results presented. What happned to the urine samples?

LIMITATIONS

Small sample size of 50 per stratum

Convenience sampling

CONCLUSION

Lines 457 and 462 ‘Contrast’ would be more appropriate than ‘contradict’

This is far too long and mixed with recommendations about poverty alleviation which is not based upon the MCA FINDINGS as presented.

Reviewers' comments:

Reviewer's Responses to Questions

**Comments to the Author**

1. If the authors have adequately addressed your comments raised in a previous round of review and you feel that this manuscript is now acceptable for publication, you may indicate that here to bypass the “Comments to the Author” section, enter your conflict of interest statement in the “Confidential to Editor” section, and submit your "Accept" recommendation.

Reviewer #1: All comments have been addressed

Reviewer #2: (No Response)

2. Is the manuscript technically sound, and do the data support the conclusions?

Reviewer #1: Yes

Reviewer #2: Partly

3. Has the statistical analysis been performed appropriately and rigorously? 

Reviewer #1: Yes

Reviewer #2: Yes

4. Have the authors made all data underlying the findings in their manuscript fully available?

Reviewer #1: Yes

Reviewer #2: Yes

5. Is the manuscript presented in an intelligible fashion and written in standard English?

Reviewer #1: Yes

Reviewer #2: No

6. Review Comments to the Author

Reviewer #1: The authors of the manuscript have addressed all the comments. They have done an excellent job, and this has improved the manuscript

Reviewer #2: 1. Please specify the contributions of each author for this research.

2. Break down the last sentence on the methods section in the abstract. It does not appear easy to read and understand.

3. Please specify which country in Lines 13 & 14 like the way it is mentioned in Line 15, “North-Western Tanzania.”

4. Lots of grammatical errors need further review.

5. Table headings are misleading. For instance, saying schistosomiasis infections may imply the various species. Please be specific.

7. PLOS authors have the option to publish the peer review history of their article (what does this mean?). If published, this will include your full peer review and any attached files.

Reviewer #1: **Yes: **Chester Kalinda

Reviewer #2: No

---

## [Author Response · Author response to Decision Letter 1]

17 Dec 2020

The title has been edited for clarity purposes

The last sentence on the methods section of the abstract has been broken down for ease of understanding

Grammatical errors corrected

Table headings have been edited so instead of schistosomiasis we have S. Mansoni

---

## [Decision Letter · Decision Letter 2]

8 Feb 2021

PONE-D-20-05468R2

Socioeconomic determinants of Schistosoma mansoni infection using multiple correspondence analysis among rural western Kenyan communities: evidence a household-based study.

PLOS ONE

Dear Dr. %ISAIAH OMONDI%,

Thank you for submitting your manuscript to PLOS ONE. After careful consideration, we feel that it has merit but does not fully meet PLOS ONE’s publication criteria as it currently stands. Therefore, we invite you to submit a revised version of the manuscript that addresses the points raised during the review process.

ACADEMIC EDITOR: Please insert comments here and delete this placeholder text when finished. Be sure to:

All the typos must be corrected,  and consistent use of acronyms and numeric symbols and spacesThe description of the choice of MCA versus alternative tools is too long and gets repeated in methods and resultsThe actual number of slides examined in unclear, and how epg was calculated and how positives by CCA only was used in the analysisThe conclusion should be concise and a fuller explanation of study limitiations would enable that conclusion.

We look forward to receiving your revised manuscript.

Kind regards,

Mary Hamer Hodges, MBBS MRCP DSc

Academic Editor

PLOS ONE

Additional Editor Comments (if provided):

You need to pay much more attention to detail and every comment made by reviewers.There are far too many typos and presentation weaknesses. Your background, methods and spill over into results augments around which method (MCA versus alternatives) to use is too long and repetitious. State your rationale for using MCA in a more straightforward manner once only. The next issue is the sampling method? The Abstract said random but the main text says convenience on more than one occasion. Please clarify how many samples were confirm positive on CCA only and how you adjusted for intensity in that instance. The conclusion need to be concise: MCA was/was not a good predictor for S. mansoni in this setting.

Reviewers' comments:

Reviewer's Responses to Questions

**Comments to the Author**

1. If the authors have adequately addressed your comments raised in a previous round of review and you feel that this manuscript is now acceptable for publication, you may indicate that here to bypass the “Comments to the Author” section, enter your conflict of interest statement in the “Confidential to Editor” section, and submit your "Accept" recommendation.

Reviewer #2: (No Response)

2. Is the manuscript technically sound, and do the data support the conclusions?

Reviewer #2: Partly

3. Has the statistical analysis been performed appropriately and rigorously? 

Reviewer #2: No

4. Have the authors made all data underlying the findings in their manuscript fully available?

Reviewer #2: Yes

5. Is the manuscript presented in an intelligible fashion and written in standard English?

Reviewer #2: No

6. Review Comments to the Author

Reviewer #2: 1. The entire paper is filled with grammatical errors, including the title.

2. What evidence do you have about schistosomiasis as a social disease (Line 10)?

3. Where is the evidence for recall bias in creating SES (line 50 & 51)? Cite the reference

4. What do you mean by this statement, “urine samples were used to investigate S. mansoni” (lines 155-159)?

5. Repeated definition of acronyms even though they have been defined earlier (Line 73).

6. Did the STH and urine data included in the analysis? If yes, report on the findings; otherwise, briefly state that STH and urine data are not reported in this paper.

7. Results interpretation is not clear. For instance, you could say S. mansoni infection was 1.42 times higher or lower in hotspot villages when compared to responder villages. You have to be consistent with the interpretation throughout the result section.

8. What do you mean by “infection prevalence” (line 320 & 337)?

9. What level of significance do you consider as “marginally significant” (lines 356-358)?

10. The entire discussion section needs to be looked at again. The section did not address the possible reasons for significant or non-significant associations with the predictor variables.

7. PLOS authors have the option to publish the peer review history of their article (what does this mean?). If published, this will include your full peer review and any attached files.

Reviewer #2: No

---

## [Author Response · Author response to Decision Letter 2]

6 Apr 2021

Grammatical errors have been rectified

Evidence of schistosomiasis as a social has been provided with a reference

Evidence for recall bias in using traditional methods of expenditure and income in generating SES has been provided

Repeated definition of acronyms has been eliminated

A statement that STH and urine results were not reported in this paper has been added

Results have been edited to enhance clarity in reporting

'infection prevalence' has been replaced with prevalence to eliminate redundancy

'marginal significance' has only been used where applicable to denote p-values>0.05 but <0.1

discussion, conclusion and limitation sections have been looked into and editing done

---

## [Decision Letter · Decision Letter 3]

26 Apr 2021

PONE-D-20-05468R3

Socioeconomic determinants of Schistosoma mansoni infection using multiple correspondence analysis among rural western Kenyan communities: evidence a household-based study.

PLOS ONE

Dear Dr. %ISAIAH OMONDI%,

Thank you for submitting your manuscript to PLOS ONE. After careful consideration, we feel that it has merit but does not fully meet PLOS ONE’s publication criteria as it currently stands. Therefore, we invite you to submit a revised version of the manuscript that addresses the points raised during the review process.

ACADEMIC EDITOR:

There remain a few areas where minor edits have been recommended

We look forward to receiving your revised manuscript.

Kind regards,

Mary Hamer Hodges, MBBS MRCP DSc

Academic Editor

PLOS ONE

Journal Requirements:

Reviewers' comments:

Reviewer's Responses to Questions

**Comments to the Author**

1. If the authors have adequately addressed your comments raised in a previous round of review and you feel that this manuscript is now acceptable for publication, you may indicate that here to bypass the “Comments to the Author” section, enter your conflict of interest statement in the “Confidential to Editor” section, and submit your "Accept" recommendation.

Reviewer #2: (No Response)

2. Is the manuscript technically sound, and do the data support the conclusions?

Reviewer #2: Yes

3. Has the statistical analysis been performed appropriately and rigorously? 

Reviewer #2: Yes

4. Have the authors made all data underlying the findings in their manuscript fully available?

Reviewer #2: Yes

5. Is the manuscript presented in an intelligible fashion and written in standard English?

Reviewer #2: Yes

6. Review Comments to the Author

Reviewer #2: Abstract

Line 20: Are you referring to “indicators” or inductors? Please correct if you are referring to indicators.

Materials and Methods

Line 161: Please if you are using “schistosomiasis” throughout the text continue to do so otherwise change “Schistosomiasis” in line 161 to “schistosomiasis”.

Line 136: Delete one “the the”

Line 303 – 304: You already defined socioeconomic status as SES please use the acronym throughout this text.

Results

Line 339: was there any significant difference between the study participants (males versus females). Please say so if any difference was observed.

Lines 357 & 358: Please rephrase this sentence “males were associated with 1:43(95% CI=1:18-1:73) times higher prevalence…..” I think it should read as “males were 1:43(95% CI=1:18-1:73) times higher prevalence compared to females”

Line 389: Again, please use the acronym (SES) for socioeconomic status

Lines 419-421: please correct the grammar in these sentences. It will read better as “….. that the EPGs among those aged below 30 was significantly higher as compared to those aged between 30-60 years and above 60 years.”

Lines 423-425: Same comment as above.

Line 427: Again, please use the acronym (SES) for socioeconomic status

Discussion

Lines 467- 470: Please define the acronyms LF, HAT, VL and NTDs you are using them for the first time.

General Comment: Please do a grammar check again especially in the area of subject verb agreement, use of prepositions and sentence structure.

7. PLOS authors have the option to publish the peer review history of their article (what does this mean?). If published, this will include your full peer review and any attached files.

Reviewer #2: No

---

## [Author Response · Author response to Decision Letter 3]

11 May 2021

In line 20, 'indicators' has been used to replace 'inductors'

the word 'schistosomiasis' has been used in lower case in all instances within the document

Most abbreviations such as 'SES' have been defined in their first occurrence and used in abbreviation elsewhere

The entire document has been reviewed for grammatical errors

---

## [Editor Report · Decision Letter 4]

28 May 2021

Socioeconomic determinants of Schistosoma mansoni infection using multiple correspondence analysis among rural western Kenyan communities: evidence from a household-based study.

PONE-D-20-05468R4

Dear Dr. %ISAIAH OMONDI%,

We’re pleased to inform you that your manuscript has been judged scientifically suitable for publication and will be formally accepted for publication once it meets all outstanding technical requirements.

Kind regards,

Mary Hamer Hodges, MBBS MRCP DSc

Academic Editor

PLOS ONE

Additional Editor Comments (optional):

Than you for your latest revision. Showing that MCA is a useful tool for assessing health disparities at the household level and that Socioeconomic status based on wealth index is not a contributing factor to schistosomiasis prevalence or intensity.
---

## [Editor Report · Acceptance letter]

2 Jun 2021

PONE-D-20-05468R4 

Socioeconomic determinants of *Schistosoma mansoni* infection using multiple correspondence analysis among rural western Kenyan communities: evidence from a household-based study.  

Dear Dr. Omondi:

I'm pleased to inform you that your manuscript has been deemed suitable for publication in PLOS ONE. Congratulations! Your manuscript is now with our production department. 

Kind regards, 

on behalf of

Dr. Mary Hamer Hodges 

Academic Editor

PLOS ONE